# A 3D-Printed Large Holding Capacity Device for Minimum Volume Cooling Vitrification of Embryos in Prolific Livestock Species

**DOI:** 10.3390/ani13050791

**Published:** 2023-02-22

**Authors:** Francisco Marco-Jiménez, Ximo Garcia-Dominguez, Luís García-Valero, José S. Vicente

**Affiliations:** Institute of Science and Animal Technology, Universitat Politècnica de València, 46022 Valencia, Spain

**Keywords:** cryopreservation, rabbit, morulae, blastocyst

## Abstract

**Simple Summary:**

Commercially available devices with simultaneous vitrification of many embryos are scarce. In this study, we developed a new three-dimensional (3D)-printed device that combines minimum volume cooling vitrification with simultaneous vitrification of a larger number of embryos. The 3D technology was stereolithography, and the Cryoeyelet^®^ device was printed in photosensitive resin. With the open Cryoeyelet^®^, 25 late rabbit morulae/early blastocysts were vitrified per device and compared with the Cryotop^®^ and the French mini-straw devices. Our results demonstrate that the CryoEyelet^®^ device can be used for the vitrification of a high number of late morulae or early blastocyst rabbit embryos per device, yielding similar outcomes to the most used commercial devices based on minimum essential volume.

**Abstract:**

Although many devices have been developed to reduce sample volume, with an explosion of methods appearing in the literature over the last decade, commercially available devices with simultaneous vitrification of a larger number of embryos are scarce, with the apparent gap for their use in prolific livestock species. In this study, we investigated the effectiveness of a new three-dimensional (3D)-printed device that combines minimum volume cooling vitrification with simultaneous vitrification of a larger number of rabbit embryos. Late morulae/early blastocysts were vitrified with the open Cryoeyelet^®^ device (*n* = 175; 25 embryos per device), the open Cryotop^®^ device (*n* = 175; 10 embryos per device), and the traditional closed French mini-straw device (*n* = 125; 25 embryos per straw) and compared in terms of in vitro development and reproductive performance after transfer to adoptive mothers. Fresh embryos constituted the control group (*n* = 125). In experiment 1, there was no difference in the development rate to the blastocyst hatching stage between the CryoEyelet^®^ and the other devices. In experiment 2, the CryoEyelet^®^ device showed a higher implantation rate compared with the Cryotop^®^ (6.3% unit of SD, *p* = 0.87) and French mini-straw^®^ (16.8% unit of SD, *p* = 1.00) devices. In terms of offspring rate, the CryoEyelet^®^ device was similar to the Cryotop^®^ device but superior to the French straw device. Regarding embryonic and fetal losses, the CryoEyelet^®^ showed lower embryonic losses compared to other vitrification devices. The analysis of bodyweight showed that all devices showed a similar outcomes—a higher birthweight but a lower body weight at puberty than those in the fresh transfer embryos group. In summary, the CryoEyelet^®^ device can be used for the vitrification of many late morulae or early blastocyst stage rabbit embryos per device. Further studies should be performed to evaluate the CryoEyelet^®^ device in other polytocous species for the simultaneous vitrification of a large number of embryos.

## 1. Introduction

Embryo cryopreservation has been a valuable tool for embryology since its first success in 1972 [1]. This technology is the best method for preserving valuable genetic resources from livestock animals [2]. Cryopreservation is also required for the widespread use of embryo transfer, which enables the exchange of genetics resources with reduced transportation cost, avoiding animal welfare problems and with minimal risk of disease transmission [3]. Undoubtedly, embryo cryopreservation is routinely used in bovine commercial embryo transfer (ET) programs. Nevertheless, commercial cryopreserved embryo transfer program for other livestock species is non-available [4]. In part, the difficulty and high cost of obtaining large numbers of embryos in these species have limited the number of cryopreservation studies performing ET transmission [3]. Nowadays, there is a considerable requirement for efficient and valuable porcine embryo cryopreservation procedures for commercial use in the swine industry as well as for genetic diversity preservation and biomedical research studies [5].

For prolific livestock species, the main handicap of vitrification is the low number of embryos that commercial devices can hold transmission [3,5,6]. Although attempts have been made to generate systems that allow cryopreservation of large numbers of embryos, such as hollow fiber [7] or the easily accessible paper container method [8], commercial, large holding capacity devices for minimum volume cooling vitrification of oocytes or embryos are non-existent. For rabbits, however, using the French straw device (without limitation on the number of embryos to be stored by device) provides an acceptable offspring production efficiency calculated as the ratio of the number of birth kits to the number of embryos transferred (ranging from 25% to 65%, [9]), while in pigs, when using the open pulled straw (OPS) method (from five to seven embryos by device) a lower offspring production efficiency has been reported (ranging from 7.1% to 23%, [9,10]. Given that 30–40 embryos are required to perform one transfer to obtain a physiological pregnancy, the ideal is to develop a system based on the minimum essential volume that allows the simultaneous vitrification of a sufficient number of embryos in a single device to carry out a single on-farm transfer [11,12]. This means that six to eight superfine open pulled-straw (SOPS) straws must be warmed to complete a single embryo transfer [6]. Recently, Gonzalez-Plaza et al. [6] demonstrated the feasibility of simultaneous vitrification of 20 morulae and blastocysts with the Cryotop^®^ device using drops with 1 to 3 embryos (0.5–1 μL each drop) that are loaded with a glass pipette stretched along the entire surface of the polypropylene sheet.

This study aimed to develop a 3D-printed device with a large holding capacity device for preimplantation embryos using the minimum volume method. Using rabbits as a model, the specific objectives were to (i) design and fabricate the device called Cryoeyelet^®^, (ii) evaluate the in vitro development efficiency, (iii) evaluate the rate of offspring, and (iv) analyze the effects on offspring phenotype compared with the Cryotop^®^ and the French mini-straw^®^ devices.

## 2. Materials and Methods

Unless otherwise noted, all chemicals were purchased from Sigma-Aldrich Qumica S.A. and were of the reagent grade (Alcobendas, Madrid, Spain).

### 2.1. Design and 3D Printing of CryoEyelet^®^ Device

The overall design goal was to create a device that could suspend a thin film within a loop with an outer cover that hermetically seals the device, optimizing space in the nitrogen tanks. The Cryoeyelet^®^ device (EP-3957715-A) is an open vitrification device consisting of a single piece (Figure 1). It includes a slender holding portion configured for holding and labeling the sample, having an intermediate portion connected to a tip and a pointed distal end comprising a hole with an elliptical configuration. Based on the preliminary trials of sample loading and vitrification feasibility, the loop (inner measures) of the vitrification component was designed with a major axis of 8.8 mm (referred to as ‘loop lengths’) and 0.6 mm of the minor axis (referred to as ‘loop widths’). With these dimensions, the device can be stored inside the straw covers. The overall length is 140 mm, which allows for storage on standard goblets in biobanks. Furthermore, the geometry allows forming a thin film of vitrification solution, where the embryos are in direct contact with the liquid nitrogen on their entire surface. This furthermore entails excellent ease of use by the handler since, once the sample is deposited on the support, no additional steps are required to reduce the volume of the cryoprotective solution. Likewise, the device provides both individual and large number storage of ova or embryos. Moreover, the holding portion of the support allows labeling with the relevant information being easy to visualize. The 3D model design was created using additive manufacturing, which reduced waste and cost.

The 3D technology is stereolithography (STL), and the files developed in design with all variations were fabricated in free cutting software (Chitubox Pro, https://www.chitubox.com/en/index) to define the printer settings and the fabrication supports. The settings were loaded into an Elego Mars Saturn 3D-printer and printed in photosensitive resin. Each device took approximately 20 min to manufacture, with a material cost of EUR 1.

### 2.2. Animals

New Zealand white rabbits were used under farm conditions. Briefly, animals were housed at the Universitat Politècnica de Valencia experimental farm in flat deck indoor cages (75 × 50 × 30 cm), with free access to water and commercial pelleted diets (minimum of 15 g of crude protein per kg of dry matter (DM), 15 g of crude fiber per kg of DM, and 10.2 MJ of digestible energy (DE) per kg of DM). The photoperiod is set to provide 16 h of light and 8 h of dark, and the room temperature is regulated to keep temperatures between 10 °C and 28 °C.

### 2.3. Collection of Embryos at the Late-Morulae Early Blastocyst Stage

Seventeen nulliparous New Zealand white does were superstimulated with a combination of follicle-stimulating hormone (FSH) (Corifollitropin alfa, 3 µg, Elonva, Merck Sharp & Dohme S.A.) and hCG (7.5 UI) [13]. Seventy-two hours after superstimulation, dams were inseminated with pooled semen from New Zealand bucks of proven fertility. Ovulation was induced with 1 µg buserelin acetate (Suprefact; Hoechst Marion Roussel, S.A., Madrid, Spain). Females were euthanized 72 h after artificial insemination, and the reproductive tract was immediately removed. Embryos were recovered by flushing each uterine horn with 10 mL Dulbecco’s phosphate-buffered saline containing 0.2% (*wt*/*vol*) bovine serum albumin (BSA). Collected embryos were counted and evaluated following the International Embryo Technology Society (IETS) criteria. Briefly, only embryos in late morulae/early blastocyst stages with homogenous cellular mass and spherical mucin coat and zona pellucida were catalogued as suitable (transferable) embryos.

### 2.4. Vitrification and Warming Procedure

Embryos were vitrified according to the methodology described by Marco-Jiménez et al. [14]. Briefly, embryos were vitrified in a two-step addition procedure. At vitrification time, embryos were transferred into an equilibration solution consisting of 10% (*vol*/*vol*) ethylene glycol and 10% (*vol*/*vol*) dimethyl sulfoxide dissolved in a base medium (BM; Dulbecco’s phosphate buffered saline (DPBS) supplemented with 0.2% [*wt*/*vol*] BSA) at room temperature (22–25 °C) for 2 min. The embryos were then transferred to a vitrification solution consisting of 20% (*vol*/*vol*) ethylene glycol and 20% (*vol*/*vol*) dimethyl sulfoxide in BM and loaded into the devices and directly plunged into LN2 within 1 min. For the CryoEyelet^®^ (*n* = 175) and French mini-straw devices (*n* = 125), a total of 25 embryos were stored in each device, while 10 embryos were stored in each Cryotop^®^ (*n* = 175). For the CryoEyelet^®^ storage, the device was placed under a microscope and the focus on the eyelet area was adjusted. The embryos were aspirated with a pipette in 2 μL of the vitrification solution. The microdrop was expelled with the embryos gently in the proximal part of the eyelet and the drop was slid towards the distal part. Thus, the film of the vitrification solution containing the embryos was distributed over the entire surface of the eyelet with a low-thickness layer.

After storage in liquid nitrogen (1 month), stored embryos were placed into CryoEyelet^®^ and Cryotop^®^ devices and were warmed by abrupt immersion of the naked devices into a Petri plate (P35) containing 0.33 M sucrose solution at 25 °C in BM for 5 min and subsequently transferred to BM solution for 5 min. The embryo warming procedure used for the French mini-straw device was based on the one-step dilution method [15]. Straws containing the embryos were removed from the liquid nitrogen and placed horizontally 10 cm from the liquid nitrogen vapor for 20–30 s. When the crystallization process began inside the mini-straw, the mini-straw was immersed in a water bath at 25 °C for 10–15 s. The mini-straw content was expelled into a plate containing 0.33 M sucrose solution at 25 °C in BM for 5 min and subsequently transferred to BM solution for 5 min. Warming embryos were scored, and only undamaged embryos were catalogued as culturable/transferable.

### 2.5. Effect of Vitrification Device on the In Vitro Development

Embryos were cultured in 500 μL of SAGE 1-Step™ HSA (CooperSurgical, Barcelona, Spain) under paraffin oil (Hypure^®^ heavy, Kitazato, Distribed, Valencia, Spain) in four well plates for 48 h at 38.5 °C in a humidified atmosphere of 5% CO_2_ in the air. After culture, embryos were evaluated morphologically under a stereomicroscope for their developmental progression until the hatching/hatched blastocyst stage.

### 2.6. Effects of the Device on the Implantation Rate, Offspring Rate at Birth and Embryonic and Fetal Losses

Between 64 and 66 h before transfer, recipient does were synchronized by intramuscular administration of 1 μg i.m. of buserelin acetate (Hoechst, Marion Roussel, Madrid, Spain). Only females that presented vulva color associated with receptive status were induced to ovulate. On the day of the embryo transfer, does were anesthetized by an i.m. injection of 4 mg/Kg of xylazine (Bayer AG, Leverkusen, Germany), followed 5–10 min later by intravenous injection into the marginal ear vein of 0.4 mL/Kg of body mass of ketamine hydrochloride (Imalgène 500, Merial SA, Lyon, France) [15]. During laparoscopy, 3 mg/kg of morphine hydrochloride (Morfina, B. Braun, Barcelona, Spain) was administered intramuscularly. Between 10–12 cryopreserved and fresh embryos were transferred per recipient (5–6 embryos into each oviduct). After transfer, does were treated with antibiotics (4 mg/kg of gentamicin every 24 h for 3 days, 10% Ganadexil, Invesa, Barcelona, Spain) and analgesics (0.03 mg/kg of buprenorphine hydrochloride, [Buprex, Esteve, Barcelona, Spain] every 12 h for 3 days and 0.2 mg/kg of meloxicam [Metacam 5 mg/mL, Norvet, Barcelona, Spain] every 24 h for 3 days).

The survival rate was assessed by laparoscopy following the previous procedure, noting implantation rate (number of implanted embryos at day 14 from total embryos transferred) and birth rate (offspring born/total embryos transferred). Embryonic losses were calculated as the difference between embryos transferred and implanted embryos. Fetal losses were calculated as the difference between total born at birth and implanted embryos.

### 2.7. Effect of Vitrification Device on Postnatal Growth Performance

Body mass differences between each progeny (CryoEyelet^®^, Cryotop^®^, French mini-straw^®^ and fresh) were assessed at birth, 4th week (weaning), and 9th week (prepubertal age).

### 2.8. Statistical Analyses

Differences between CryoEyelet^®^ and other vitrification devices (Cryotop^®^ and French mini-straw^®^) and the fresh group were estimated using Bayesian inference. The dendrogram obtained by Bayesian interference was created by 60,000 interactions of Markov chain Monte Carlo, with a burn-in period of 10,000, saving only 1 of every 10 samples for inference. The parameters obtained from the marginal posterior distributions of the relative abundance between groups were the mean of the difference (DCryoEyelet^®^-j; computed as CryoEyelet^®^-j, being j the Cryotop^®^, French mini-straw^®^ and fresh groups), the probability of the difference being greater than 0 when DCryoEyelet^®^-j > 0 or lower than 0 when DCryoEyelet^®^-j < 0 (P0), and the highest posterior density region at 95% of probability (HPD95%). Di-j estimated the mean of the differences between i and j traits, P0 estimated the probability of DCryoEyelet^®^-j ≠ 0, and HPD95% estimated the accuracy. Statistical difference was considered if P0 > 0.8 (80%). Statistical analysis was computed with the rabbit program developed by the Institute for Animal Science and Technology (Valencia, Spain).

## 3. Results

A total of 600 embryos were utilized, from which 225 were used to evaluate the in vitro hatching rates (in vitro assay distributed in each group as follows: CryoEyelet^®^
*n* = 75, Cryotop^®^
*n* = 75, French Straw^®^
*n* = 50, and Fresh *n* = 20), and 375 were transferred to foster mothers (*n* = 37) to evaluate the offspring rate (distributed in each group as follows: CryoEyelet^®^
*n* = 100, Cryotop^®^
*n* = 100, French Straw^®^
*n* = 100, and Fresh *n* = 75). Descriptive data of traits are annotated in Table 1.

In experiment 1, there was no difference in the development rate to the blastocyst hatching stage between CryoEyelet^®^ and the other devices (Table 2), but CryoEyelet^®^ results slightly decreased compared to the fresh group (−6.3% unit of SD, *p* = 0.80, Table 2). In experiment 2, CryoEyelet^®^ showed a higher implantation rate compared with Cryotop^®^ (6.3% unit of SD, *p* = 0.87, Table 2) and French mini-straw^®^ devices (16.8% unit of SD, *p* = 1.00, Table 2). There was no difference between CryoEyelet^®^ and the fresh group (Table 2). In terms of offspring rate, CryoEyelet^®^ was similar to Cryotop^®^ (Table 2) but superior to the French straw device. As expected, the results of the CryoEyelet^®^ device were inferior to the fresh group (−13.2% SD unit, *p* = 0.97, Table 2). Regarding embryonic and fetal losses, CryoEyelet^®^ showed lower embryonic losses compared to other vitrification devices (Table 2), while fetal losses were the same as with the Cryotop^®^ device but again lower than with the French straw device (Table 2). The embryonic losses were similar to the fresh group but higher in fetal losses (Table 2).

At birth, offspring derived by the CryoEyelet^®^ device exhibited a higher bodyweight compared to the other devices and the fresh group (using the litter size covariate 8.96, significant effect at *p* < 0.05, Table 3). At weaning (4 weeks old), kits derived from CryoEyelet^®^ showed a similar bodyweight to the Cryotop^®^ device but a higher bodyweight compared to the French straw devices (55.4 g SD unit, *p* = 0.92, Table 3). At rearing (9 weeks old), animals derived from CryoEyelet^®^ exhibited similar bodyweight to the other devices (Table 3) but were higher than the fresh group (−132.2 g SD unit, *p* = 0.91, Table 3).

## 4. Discussion

In this study, we showed a simple, low-cost, 3-D printable, practical vitrification device based on the minimum volume method for the simultaneous vitrification of a large number of rabbit embryos. Our findings revealed that the CryoEyelet^®^ is a suitable system for the simultaneous vitrification of almost 25 rabbit embryos at the late morula/early blastocyst stages, with efficiency in terms of in vitro and in vivo development similar to the Cryotop^®^ device, considered one of the gold standards for human gamete cryopreservation [16,17].

Although a great number of devices have been developed to reduce sample volume over the last decade [18,19], with some of the cryodevices being commercially available (e.g., Cryotop^®^/Cryotop^®^CL, Cryolock^®^, CryoTip^®^, and Diamour-op/Diamour-cs), devices for the simultaneous vitrification of a large number of embryos are scarce, with the apparent gap for their use in prolific livestock species [3,6]. To vitrify a large number of embryos not only greatly simplifies the current vitrification protocols in prolific livestock species but also facilitates the embryo warming and embryo transfer processes, generally performed under field conditions (one device/one transfer, [6]).

Three-dimensional (3-D) printing and computer-aided design (CAD) offer practical ways to quickly design and construct devices that can support cryogenic applications due to their fast expanding consumer-level capabilities [20,21]. Thus, the work presented herein produced a 3D-printed vitrification device based on the cryoloop procedure [22,23] that can store almost 25 rabbit preimplantational embryos and can be accommodated in 0.25 mL straws for space optimization during storage. With the CryoEyelet^®^ device, there was no difference in the ability of preimplantational embryos to hatch in vitro. Vitrification using the French mini straw^®^ [24] and Cryotop^®^ [14] devices and the loop procedure (calibrated plastic inoculation loop, [18]) were previously used to successfully vitrify late-morulae/early-blastocyst stage rabbit embryos. Similar to this study, the in vitro viability of rabbit embryos vitrified at the late-morulae/early-blastocyst stage does not appear to be affected by the device used [14,25]. Thus, the CryoEyelet^®^, Cryotop^®^, and French mini-straw^®^ devices show a similar and excellent in vitro development capacity. Regardless of the device used, the in vitro results were close to the fresh group.

The ultimate test of the viability of embryos after cryopreservation is the ability to establish and maintain a pregnancy, resulting in normal fertile young [23]. In this study, the implantation rates of rabbit late-morulae/early-blastocysts vitrified with the CryoEyelet^®^ were higher than that of the Cryotop^®^ (+6.3%) and French mini-straw^®^ (+16.8%) embryos. Moreover, the implantation rates of vitrified embryos with the CryoEyelet^®^ were similar to that of fresh embryos. In terms of offspring rate, CryoEyelet^®^ was similar to Cryotop^®^ but superior to the French straw (8.5%) device. Until now, the Cryotop device has provided the most successful rate of offspring at birth in rabbits [14,25]. However, the Cryotop vitrification device is expensive for livestock applications and conservation-focused biobanks, with an average cost of USD 20–30 per sample, which only seems justifiable for human fertility clinics. In addition, the Cryotop^®^ system typically stores only 1–4 oocytes or embryos at a time, a process which is performed by skilled embryologists. In our study, 10 embryos were successfully placed and vitrified in each Cryotop^®^ device according to the manufacturer’s instructions. Recently, Gonzalez-Plaza et al. [6] demonstrated that Cryotop^®^ is suitable for the simultaneous vitrification of at least 20 porcine embryos at the morula or blastocyst stage by forming steric droplets (groups of 1–3 embryos in 0.5–1 μL) along the polypropylene sheet. A notable advantage of the CryoEyelet^®^ device is its ease of use, making it accessible to embryologists unfamiliar with the Cryotop^®^ device. Additionally noteworthy is that the CryoEyelet^®^ device is considerably lower in cost (the retail cost could be roughly USD 3 per sample) compared with the commercial vitrification devices based on minimum essential volume.

Due to this reason, multi-ovulation and embryo-transfer (MOET) intervention appear to influence measurable outcomes of offspring physiology, manifesting differently across the species studied [26], including rabbits [25]; we investigated the impact of the vitrification device on postnatal bodyweights. Previously, we reported that animals born combined MOET with cryopreservation exhibit higher birthweight and poor growth performance independently of the vitrification device using a rabbit model [25]. Newly, here we demonstrated that animals born after embryo cryopreservation exhibit higher birthweight and poor growth performance independently of the device used, in line with several studies in different mammalian species, including humans [26,27,28,29]. Importantly, it can be noted that these measurable differences are noticeable in healthy, fertile animal populations [25]. All devices show a similar outcome concerning the bodyweight, a higher birthweight but a lower body weight at puberty than those in the fresh transfer embryos group. The main limitation of our results is related to the limited literature that compared offspring outcomes following vitrification compared several devices. Given the mounting evidence from both animal and human research that offspring born after the use of ART may exhibit physiologic alterations from those who are spontaneously conceived, more significant consideration of the vitrification devices used for more precise decision-making regarding the application of this technology [26]. Further studies would be of interest, as they allow these effects to be studied in other species.

## 5. Conclusions

In summary, the CryoEyelet^®^ device can be used for the vitrification of a high number of late morulae or early blastocyst rabbit embryos per device, yielding similar outcomes to the most used commercial device based on minimum essential volume. Further studies should be performed to evaluate the successful vitrification with the CryoEyelet^®^ device with many embryos in other polytocous species.

## Figures and Tables

**Figure 1 animals-13-00791-f001:**
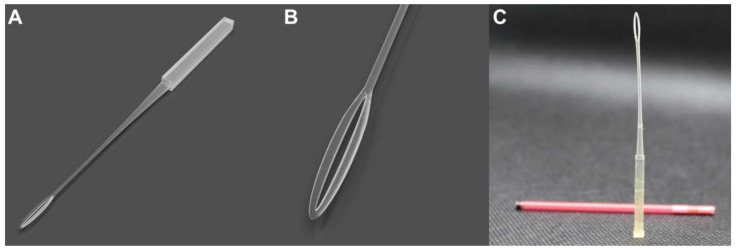
Vitrification device design and 3D-printing prototype. (**A**) Full device rendering. (**B**) Detail of the gamete and embryo deposition area. (**C**) 3D-printed device with a cap to generate a closed system.

**Table 1 animals-13-00791-t001:** Summary of the traits and means from each experimental group analyzed.

Trait	CryoEyelet^®^	Cryotop^®^	French Straw^®^	Fresh
In vitro development (%)	89.3	88.3	90.2	95.4
Implantation rate (%)	84.3	78.0	67.5	87.5
Offspring rate (%)	60.7	65.1	52.3	73.9
Losses (%)				
Embryonic	15.5	21.9	32.5	12.5
Fetal	30.3	31.0	22.3	15.5
Bodyweight (g)				
At birth	64.7	54.5	53.6	47.7
At weaning	513.0	523.2	457.6	498.8
At prepubertal age	1560.1	1521.5	1692.3	1547.9

**Table 2 animals-13-00791-t002:** Embryo viability rates differences using CryoEyelet^®^, Cryotop^®^, and French Straw^®^ devices and a fresh control group. Data computed as CryoEyelet^®^−Cryotop^®^, CryoEyelet^®^−Straw^®^ and CryoEyelet^®^−Fresh.

Trait	Device Comparisons								
	CryoEyelet^®^−Cryotop^®^			CryoEyelet^®^−French Straw^®^			CryoEyelet^®^−Fresh		
	Di-j	P0	HPD95%	Di-j	P0	HPD95%	Di-j	P0	HPD95%
In vitro development (%)	7.8	0.56	−9.67, 11.36	−1.1	0.58	−12.22, 9.25	−6.3	0.80	−22.28, 7.54
Implantation (%)	6.3	0.87	−4.44, 17.64	16.8	1.00	5.78, 28.49	−3.2	0.72	−15.27, 7.55
Offspring rate (%)	−4.4	0.75	−17.54, 8.79	8.5	0.89	−5.12, 22.19	−13.2	0.97	−27.29, −0.18
Losses (%)									
Embryonic	−6.4	0.88	−17.62, 4.05	−0.2	1.00	−27.94, −5.29	30.9	0.71	−8.01, 13.84
Fetal	−0.7	0.55	−14.16, 11.26	7.9	0.86	−6.98, 21.85	14.8	0.99	1.61, 27.37

Di-j = mean of the difference between i–j devices (median of the marginal posterior distribution of the difference between the i and j devices); P0 = probability of the difference (Di-j) being greater than 0 when i-j > 0 or lower than 0 when Di-j < 0; HPD95% = the highest posterior density region at 95% of probability. Statistical differences were assumed if P0 > 0.80.

**Table 3 animals-13-00791-t003:** Bodyweight progenies differences using CryoEyelet^®^, Cryotop^®^, and French Straw^®^ devices and fresh control group. Data computed as CryoEyelet^®^−Cryotop^®^, CryoEyelet^®^−Straw^®^ and CryoEyelet^®^−/Fresh.

Bodyweight (g)	Device Comparisons								
	CryoEyelet^®^−Cryotop^®^			CryoEyelet^®^−French Straw^®^			CryoEyelet^®^−Fresh		
	Di-j	P0	HPD95%	Di-j	P0	HPD95%	Di-j	P0	HPD95%
At birth	10.3	1.00	5.06, 15.48	11.1	1.00	6.14,16.25	16.9	1.00	10.09, 23.57
At weaning	−10.2	0.60	−94.34, 70.54	55.4	0.92	−23.76, 133.60	14.2	0.60	−101.89, 117.94
At rearing	38.6	0.70	−107.49, 183.91	12.2	0.57	−126.80, 153.04	−132.2	0.91	−326.06, 68.83

Di-j = mean of the difference between i–j devices (median of the marginal posterior distribution of the difference between the i and j devices); P0 = probability of the difference (Di-j) being greater than 0 when i-j > 0 or lower than 0 when Di-j < 0; HPD95% = the highest posterior density region at 95% of probability. Statistical differences were assumed if P0 > 0.80.

## Data Availability

The original contributions presented in the study are included in the article/supplementary material, and further inquiries can be directed to the corresponding authors.

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
