# Peer review of "A 3D-Printed Large Holding Capacity Device for Minimum Volume Cooling Vitrification of Embryos in Prolific Livestock Species"

_animals, 2023, doi:10.3390/ani13050791_

Round 1

Reviewer 1 Report

This manuscript is well written, the subject is within the scope of the journal, the experimental design is over all adequate and the results provide interesting information regarding the possibility of a new 3-d printed device that can be used for the vitrification of a high number of rabbit embryos per device. There are, however, some aspects that needed to be better explained:

-       In page 3: “Seven nulliparous New Zealand White does were super-stimulated … Females were euthanised 72 h after artificial insemination”. Why wasn’t the laparoscopy method used here? 1 super stimulation/doe in 7 does resulted in 600 embryos?

-       In page 4: “The embryos were then transferred to a vitrification solution … and loaded into the devices.” For the new 3d printed device it would be important to give details on how loading was done, as this is the 1stdescription of its use.

-       In page 4: “After storage in liquid nitrogen…” How long were embryos stored before use?

-       Number of embryos per group (vitrified per device and fresh) is only indicated in the abstract, but not in M&M and it should be.

-       In page 4: “Laparoscopically embryo transfers were performed on 64–66 hours of the oestrous cycle [15]. Their ovulation was induced in receptive…” – These sentences need rephrasing.

-       In page 4: “Between 10–12 cryopreserved and fresh embryos were transferred per recipient (5-6 embryos into each oviduct)”. How many recipients per group?

-       In page 5: “A total of 600 embryos were utilised, from which 225 were used to evaluate the in vitro hatching rates (in vitro assay), and 375 were transferred to foster mothers to evaluate the offspring rate (n=12).” These refer to total number used, it should be clear in text how many embryos (for in vitro and for transfer per group) and recipients were used per group. 

Author Response

Dear reviewer,

Thank you very much for your comments which will help to improve the quality of this work. 

-       In page 3: “Seven nulliparous New Zealand White does were super-stimulated … Females were euthanised 72 h after artificial insemination”. Why wasn’t the laparoscopy method used here? 1 super stimulation/doe in 7 does resulted in 600 embryos?

This is a mistake as there are 17 females, not 7 (average number of embryos per female 35 embryos). Indeed, it is sometimes possible to use the laparoscopic technique to recover embryos in vivo, but this is only used for valuable animals (e.g. GMOs) due to its economic cost.

-       In page 4: “The embryos were then transferred to a vitrification solution … and loaded into the devices.” For the new 3d printed device it would be important to give details on how loading was done, as this is the 1stdescription of its use.

A new sentences have been added: “For the CryoEyelet® storage, place the device under a microscope and adjust the focus on the eyelet area. Aspirate the embryos with a pipette in 2 μL of the vitrification solution. Expelled the microdrop with the embryos gently in the proximal part of the eyelet and slid the drop towards the distal part. Thus, the film of vitrification solution containing the embryos is distributed over the entire surface of the eyelet with a low-thickness layer”

-       In page 4: “After storage in liquid nitrogen…” How long were embryos stored before use?

The embryos were stored for 1 month. This information has been included in the text.

-       Number of embryos per group (vitrified per device and fresh) is only indicated in the abstract, but not in M&M and it should be.

The number of embryos vitrified for each device have been included in M&M.151

-       In page 4: “Laparoscopically embryo transfers were performed on 64–66 hours of the oestrous cycle [15]. Their ovulation was induced in receptive…” – These sentences need rephrasing.

The sentences have been rephrasing

-       In page 4: “Between 10–12 cryopreserved and fresh embryos were transferred per recipient (5-6 embryos into each oviduct)”. How many recipients per group?

A total of 37 receivers were used, distributed as follows: CryoEyelet® (n=10), Cryotop (n=10), Straw (n=10) and fresh group (n=7).

-       In page 5: “A total of 600 embryos were utilised, from which 225 were used to evaluate the in vitro hatching rates (in vitro assay), and 375 were transferred to foster mothers to evaluate the offspring rate (n=12).” These refer to total number used, it should be clear in text how many embryos (for in vitro and for transfer per group) and recipients were used per group. 

We fully understand the reviewer's point concerning the numbers that were not correct for the receivers used. A new sentences have been added describing the data by group for both the in vitro and in vivo experiments

Reviewer 2 Report

The authors compared a new device produced by 3D-printing with two commercially available devices suitable for vitrification of several embryos in the same device. Rabbit embryos collected from superovulated does were used. Their survival after in vitro culture as well as their outcome after transfer in recipient does were evaluated up to the weaning of the offspring.  The authors demonstrated that their new device allows to obtain similar or sometimes better results than the two other devices.

The study is original and seems well conducted. The proposed device is cheaper than those on the market and seems easy to handle. The text is in general easy to understand.

Specific comments

It is mentioned in the introduction that the main goal is to develop a device for vitrification of pig embryos. Rabbit and pig embryos differ by several criteria, including their fat contents, which is known to impact their sensitivity to cryopreservation and the importance of the rapidity of cooling. Moreover, important differences are also observed between in vitro and in vivo produced embryos (the former being much more sensitive). A short paragraph mentioning those differences could be added in the discussion.

Experimental design, statistical analysis and results:

·       1.   It is not clear if embryos cryopreserved in different conditions were transferred in each horn, so that each recipient receives embryos from two different conditions, which might improve the power of the analysis by taking into account the “recipient” effect. More information on the experimental design would help.

·       2.   I am not familiar with bayesian inference. So I do not understand how the authors determine the likelihood of a difference between treatments and how they determine the threshold for a significant difference between conditions. If I understood properly, it has been put at P0>0.8, which seems equivalent to a p<0.2 and not to a threshold of p<0.05 widely used with “classical” statistics. Why not instead using the HPD95% and consider the difference to be significant if the 0 is not included in this interval?  

·       3.  To improve the visibility of the data, I suggest to replace the table 1 by a figure (box plots).

·        4.  The number (proportion) of embryos found degenerated at warming (before culture or transfer)is not mentioned.

·       5.   Table 1 should include the number of embryos/foetus/newborns at each stage for each treatment (if a figure is used, this information has to be given in the legend or in the text).

·     6.     Title of the tables: what means “identified by researchers”?

·     7.  Is the material used to produce the CryoEyelet device suitable for long term conservation in liquid nitrogen? How is it sterilized? Can it be reused?

Author Response

Dear reviewer,

Thank you very much for your comments which will help to improve the quality of this work. 

Specific comments

It is mentioned in the introduction that the main goal is to develop a device for vitrification of pig embryos. Rabbit and pig embryos differ by several criteria, including their fat contents, which is known to impact their sensitivity to cryopreservation and the importance of the rapidity of cooling. Moreover, important differences are also observed between in vitro and in vivo produced embryos (the former being much more sensitive). A short paragraph mentioning those differences could be added in the discussion.

The references to the pig model throughout the study are simply intended to offer the potential use of this new device. Moreover, due to the peculiarity of pig embryos (lipids content), we considered that only when a test is performed using these embryos will we be able to reveal the potential for use. In the meantime, incorporating a short paragraph mentioning those differences would be highly speculative, and we do not consider it appropriate.

Experimental design, statistical analysis and results: 

 It is not clear if embryos cryopreserved in different conditions were transferred in each horn, so that each recipient receives embryos from two different conditions, which might improve the power of the analysis by taking into account the “recipient” effect. More information on the experimental design would help.

Each recipient only received embryos from one experimental group.  The use of each uterine horn with one type of embryos would require caesarean surgery, which is not advisable in this experiments for animal welfare reasons. Moreover, the recipient effect (variability) is partly reduced by using 10 recipient animals per group.

I am not familiar with bayesian inference. So I do not understand how the authors determine the likelihood of a difference between treatments and how they determine the threshold for a significant difference between conditions. If I understood properly, it has been put at P0>0.8, which seems equivalent to a p<0.2 and not to a threshold of p<0.05 widely used with “classical” statistics. Why not instead using the HPD95% and consider the difference to be significant if the 0 is not included in this interval?  

This is an entirely different approach to frequentist statistics. Bayesian inference is a method of statistical inference in which Bayes' theorem is used to update the likelihood of a hypothesis as more data become available. The likelihood is multiplied by an a priori distribution, and random draws from this product; the a posteriori distribution usually summarises inferences. At its core, frequentist statistics is about repeatability and gathering more data. The frequentist interpretation of probability is the long-run frequency of repeatable experiments, while in the Bayesian, probability represents our degree of belief in something, which is probably closer to most people's intuitive idea of probability.

 Bayesian inference is very useful when a low number of data is available, and the expected differences between groups can be low (e.g. 10%). Trying to observe differences with the frequentist statistic implies many animals, which goes against the 3R principles.

To improve the visibility of the data, I suggest to replace the table 1 by a figure (box plots).

The transformation of table 1 into a figure is not recommended as it would imply generating 8 independent figures with 4 groups per figure. It would not be straightforward to visualise. We consider it more suitable to keep the description in numerical format.

The number (proportion) of embryos found degenerated at warming (before culture or transfer) is not mentioned.

Morphoanomalous embryos are discarded after recovery of the donor females. Once vitrification is used on these embryos, in rabbits, it is usual that no degenerated embryos are observed after devitrification.

Table 1 should include the number of embryos/foetus/newborns at each stage for each treatment (if a figure is used, this information has to be given in the legend or in the text).

Title of the tables: what means “identified by researchers”?

The term identified by researchers has been removed.

 Is the material used to produce the CryoEyelet device suitable for long term conservation in liquid nitrogen? How is it sterilized? Can it be reused?

Sterilisation was carried out by placing the devices under ultraviolet light for 20 minutes. I do not recommend reusing it at this time. However, there are materials such as Recreus PP3D (medical degree) that can be autoclaved; therefore, it would be possible to consider reuse. We will have to test this.

Reviewer 3 Report

This is an interesting paper and indicating a new innovation in vitrification techniques.  I do not have any issue with the number embryos loaded into each device, because it became a common practice in the facility where the cost for the device cannot be passed on to the patients.  Also the result is promising.  However there are few issue that need to be addressed by the author.  

1.  Material method: there is no explanation for the tools/equipment to handle a high number embryos during vitrification and warming processes.

2. Volume of the drop in each device, especially for the CryoEyelet.  Since to handle up to 25 embryos.

3.  Volume and sealing system for straw.

The volume used for vitrification should also be reflected in the discussion since most vitrification devices indicated the smallest volume to obtain the best result.

regards, 

Author Response

Dear reviewer,

Thank you very much for your comments which will help to improve the quality of this work. 

  1. Material method: there is no explanation for the tools/equipment to handle a high number embryos during vitrification and warming processes.

Our device allows manipulation with a micropipette. No unique system is needed. It is possible to use a stripper pipette or a handmade glass pipette. A better description of how embryos are deposited in the device has been included.

  1. Volume of the drop in each device, especially for the CryoEyelet.  Since to handle up to 25 embryos.

A paragraph has been included to explain the handling of the new device better. For the CryoEyelet® storage, place the device under a microscope and adjust the focus on the eyelet area. Aspirate the embryos with a pipette in 2 μL of the vitrification solution. Expelled the microdrop with the embryos gently in the proximal part of the eyelet and slid the drop towards the distal part. Thus, the film of vitrification solution containing the embryos is distributed over the entire surface of the eyelet with a low-thickness layer

  1. Volume and sealing system for straw.

With rabbit embryos, vitrification is done in 100 μl of vitrification solution when the straw device is used. The loading is similar to the use of a slow freezing procedure.

Reviewer 4 Report

animals-2196950

“A 3D-printed large holding capacity device for minimum volume cooling vitrification of embryos in prolific livestock specie”

In the present work, authors developed a new device for vitrification of a large number of embryos. The study gives answer to an old problem related to the difficulty of finding an appropriate device that efficiently accommodates a large number of embryos, making the interventions more efficient and sustainable. The approach is correct, the experiment design is complete and manuscript is well-written. The number of ET and embryo evaluation is important and sufficiently robust in order to allow reliable conclusions. I have minor points.

Pg 2- Please replace reference Marco-Jimenez et al, 2018 by the correspondent number.

Pg 2- Please revise the expression “live birth kits”., i.e., if it is appropriate in the context.

-Please write firstly the words unabbreviated before the abbreviated. This applies to OPS and SOPS, Superfine Open Pulled Straw, for instance.

-Systems that allow direct contact between the vitrification solution and nitrogen expose material to biological risk. How can this be avoided with the present device?

-How long were the embryos kept preserved in liquid nitrogen before thawing?

-How many recipient females were used in the present study?

-A stereomicroscope couldn’t be sophisticated enough to evaluate embryos accurately in terms of embryonic cells number and other aspects. Why did authors opted by this method of evaluation?

-Difference between implantation rate and birth rate i.e. embryonic and foetal losses could be due to other causes then the embryo quality in all tested groups (even with the administration of antibiotics post-transfer). How did authors control this aspect?

Author Response

Dear reviewer,

Thank you very much for your comments which will help to improve the quality of this work. 

Pg 2- Please replace reference Marco-Jimenez et al, 2018 by the correspondent number. 

Correction has been made

Pg 2- Please revise the expression “live birth kits”., i.e., if it is appropriate in the context.

Correction has been made

-Please write firstly the words unabbreviated before the abbreviated. This applies to OPS and SOPS, Superfine Open Pulled Straw, for instance. 

Correction has been made

-Systems that allow direct contact between the vitrification solution and nitrogen expose material to biological risk. How can this be avoided with the present device? 

Like all other open systems, this device must be used in a single vessel with sterile nitrogen. Each vitrification vessel must be sterilised after use. In future studies, we will evaluate its response as a closed system.

-How long were the embryos kept preserved in liquid nitrogen before thawing?

They were stored for 1 month. This data has been included in M&M

-How many recipient females were used in the present study? 

A total of 37 recipients were used. This data has been included in the results section

-A stereomicroscope couldn’t be sophisticated enough to evaluate embryos accurately in terms of embryonic cells number and other aspects. Why did authors opted by this method of evaluation?

We use in vitro evaluation as a routine technique to rule out a priori systems that clearly will not generate offspring. Our results are based on the ability to develop to blastocyst, and this is feasible using a stereomicroscope with 11.5X magnification. In our experiments, the final examination is whether these embryos will generate offspring (according to the baby born at the home concept).

-Difference between implantation rate and birth rate i.e. embryonic and foetal losses could be due to other causes then the embryo quality in all tested groups (even with the administration of antibiotics post-transfer). How did authors control this aspect?

It has not been observed in the rabbit model, which, together with our extensive experience in embryo transfer in this species, makes us not contemplate this effect. However, in all our animals, the treatment received is the same, and there is no interaction of this effect.